# Ultrasonic activation of inert poly (tetrafluoroethylene) enables piezocatalytic generation of reactive oxygen species

Yanfeng Wang[1], Yeming Xu[2], Shangshang Dong[1], Peng Wang [2], Wei Chen[3], Zhenda Lu [2], Deju Ye [4], Bingcai Pan [1,5], Di Wu [2], Chad D. Vecitis [6] & Guandao Gao [1,5✉]

Controlled generation of reactive oxygen species (ROS) is essential in biological, chemical, and environmental fields, and piezoelectric catalysis is an emerging method to generate ROS, especially in sonodynamic therapy due to its high tissue penetrability, directed orientation, and ability to trigger in situ ROS generation. However, due to the low piezoelectric coefficient, and environmental safety and chemical stability concerns of current piezoelectric ROS catalysts, novel piezoelectric materials are urgently needed. Here, we demonstrate a method to induce polarization of inert poly(tetrafluoroethylene) (PTFE) particles (<d> ~ 1–5 μm) into piezoelectric electrets with a mild and convenient ultrasound process. Continued ultrasonic irradiation of the PTFE electrets generates ROS including hydroxyl radicals ($\bullet OH$), superoxide ($\bullet O_2^-$) and singlet oxygen ($^1O_2$) at rates significantly faster than previously reported piezoelectric catalysts. In summary, ultrasonic activation of inert PTFE particles is a simple method to induce permanent PTFE polarization and to piezocatalytically generate aqueous ROS that is desirable in a wide-range of applications from environmental pollution control to biomedical therapy.

[1] State Key Laboratory of Pollution Control and Resource Reuse, School of Environment, Nanjing University, Nanjing, China. [2] National Laboratory of Solid Microstructures, Department of Materials Science and Engineering, College of Engineering and Applied Sciences, and Collaborative Innovation Center of Advanced Microstructures, Nanjing University, Nanjing, China. [3] College of Environmental Science and Engineering, Ministry of Education Key Laboratory of Pollution Processes and Environmental Criteria, Tianjin Key Laboratory of Environmental Remediation and Pollution Control, Nankai University, Tianjin, China. [4] State Key Laboratory of Analytical Chemistry for Life Science, School of Chemistry and Chemical Engineering, Nanjing University, Nanjing, China. [5] Research Center for Environmental Nanotechnology (ReCENT), Nanjing University, Nanjing, China. [6] John A. Paulson School of Engineering and Applied Sciences, Harvard University, Cambridge, MA, USA. ✉email: gaoguandao@nju.edu.cn

Reactive oxygen species (ROS), such as hydroxyl radicals (•OH), superoxide (•$O_2^-$), and singlet oxygen ($^1O_2$), are among the strongest aqueous redox species, and their effective and efficient production is desired in biological, chemical, and environmental fields[1–4]. Recently, piezocatalysis has been demonstrated as a new advanced oxidation process where low-frequency vibration or high-frequency ultrasound waves induce the polarization and establish built-in electric field in piezocatalysts, resultantly, electrons and holes can be continuously separated and attracted on the opposite surface for piezocatalytic redox reactions[5–9]. In an aqueous solution, these surficial charge carriers can undergo oxidation and reduction reactions with water or dissolved species yielding homogeneous ROS that may have a range of applications. For example, piezocatalytic ROS generation may find utility in sonodynamic therapy due to its high tissue penetrability, directed orientation, and ability to trigger in situ ROS generation[10–14].

Piezocatalytic ROS generation efficiency is by nature dependent on piezoelectric coefficient, $d_{33}$, and generally the piezocatalytic activity increases with increasing $d_{33}$. Classical piezoelectric materials that have been demonstrated as piezocatalysts include inorganic $BaTiO_3$, ZnO, and $BiFeO_3$ as well as organic polyvinylidene fluoride (PVDF)[15–21]. However, the piezoelectric coefficient ($d_{33} \approx 3$–105 pC/N) is too low to be effective for piezocatalysis applications. Lead zirconate titanate (PZT) has a reasonable $d_{33}$ (265 pC/N) for piezocatalysis, but lead has environmental and human health issues and friendly and is unstable under ROS generating conditions[22,23].

Nonpolar polymer electret materials such as poly(tetrafluoroethylene) (PTFE; Teflon), polypropylene, and polystyrene are dielectrics that can quasi-permanently store charge or polarization. These organic electrets have been widely utilized in transducers (e.g., microphones and loudspeakers), electrophotography, electroactive air filters, and generators[24]. Meanwhile, polymeric electrets have also been reported to have large apparent piezoelectric coefficients, which can be more than an order of magnitude greater than that of conventional piezoelectric polymers (e.g., polyvinylidene fluoride) and approach the highest values of well-studied inorganic piezoelectric materials[25–27]. For example, the piezoelectric $d_{33}$ coefficients of the PTFE electret was reported to reach ~600 pC/N, higher than that of traditional piezoelectric materials[25]. Generally, PTFE is known to be extremely inert even under the most harsh conditions such as strong acids, strong alkalis, and several hundred solvents below 300 °C[28]. In comparison, the PTFE electret has some chemical activity as it has been applied in the dust removal industry in the form of a charged filter membrane that utilizes the electrostatic attraction between charged PTFE electret and the charged dust particles[29]. However, the traditional PTFE polarization method is complex, typically involving a high-voltage electric-field polarization in the range of 200 MV/m[30,31]. Thus, simple methods to active PTFE electret particles are needed to enable future fundamental studies on PTFE electret piezocatalysts and their potential applications.

Herein, we report on a method to induce polarization of inert PTFE particles (average 1–5 μm) (Supplementary Fig.1) to a piezoelectric electret via a mild and convenient ultrasonic irradiation process. Continued ultrasonication of the PTFE electrets drives piezoelectric generation of aqueous ROS (Fig. 1) at rates significantly greater than previously reported piezoelectric catalysts. The fundamental mechanisms involved in the ultrasonic activation of inert PTFE as well as the PTFE electret piezocatalytic generation of ROS were investigated by piezoresponsive force microscopy (PFM) and electron spin resonance (ESR), respectively. The potential environmental and biomedical applications of the stable PTFE piezocatalyst are discussed.

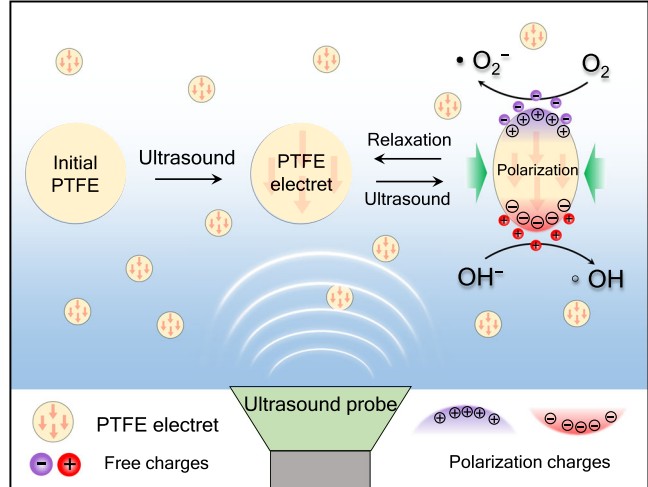

**Fig. 1 Schematic illustration of the piezocatalytic generation of ROS by PTFE under ultrasound irradiation.** The initial PTFE was first activated to become a permanent polarized piezoelectric PTFE electret under ultrasonic irradiation. Then the charges from the surrounding electrolyte were adsorbed on the surface of PTFE. Subsequently, the adsorption of charges will be released as free charges under compressive stress and interact with water molecules or oxygen to produce homogeneous ROS.

## Results and discussion

**Physical mechanism on PTFE activation.** PTFE particles or membranes were first activated under ultrasonic irradiation (40 kHz, 110 W), then PFM was used to examine induced PTFE piezoelectric properties[32]. Notably, the PFM phase contrast and amplitude response of PTFE were sharply enhanced with a nearly 120° phase change occurring between neighboring micron-sized domains after the PTFE was ultrasonically irradiated (Supplementary Fig. 2). This indicates the emergence of PTFE electret domains with strong localized piezoelectric properties. Ultrasonic pressure waves force the formation and quasi-adiabatic collapse of vapor bubbles formed from pre-existing gas nuclei. The transient collapse of acoustic cavitation bubbles can generate extremely high pressures (~100 Mpa) and electric fields (~100 kV/m)[33,34]. These transient and extreme ultrasonic cavitation pressure waves can cause massive PTFE deformation resulting in permanent structural defects. As a consequence, the concurrent electric fields can generate charges that can be trapped in the PTFE structural defects creating an electret state[26]. In addition, although the initial PTFE as a whole is centrosymmetric, some of the local regions near these chemical or physical defects may cause PTFE to be in a localized noncentrosymmetric phase[35]. In agreement with experiments, recent theoretical modeling has reported the piezoelectric behavior of the PTFE electret to be due to the presence of charges, the interaction of Maxwell stress, and deformation nonlinearity[36,37]. To test this hypothesis, we attempted to polarize the PTFE by applying a high pressure using a tableting machine (25 Mpa) and applying a strong electric field (134 kV/m). The PFM phase images of the pressure and electric field treated PTFE display similar localized micron-scale phase shifts, the ultrasound treated PTFE, especially when compared to the untreated PTFE that is spatially homogeneous in phase (Fig. 2a, b). In addition, the piezoelectric amplitude of PTFE (154.7 ± 77.8 pm) measured by PFM was 23.8 times higher than that of classical PVDF piezoelectric (6.5 ± 1.2 pm) (Fig. 2c). According to the equation $A = d_{33}V_{ac}Q$, where $A$, $d_{33}$, $V_{ac}$, and $Q$ are piezoelectric amplitude, piezoelectric coefficient, AC voltage applied to the specimen through the

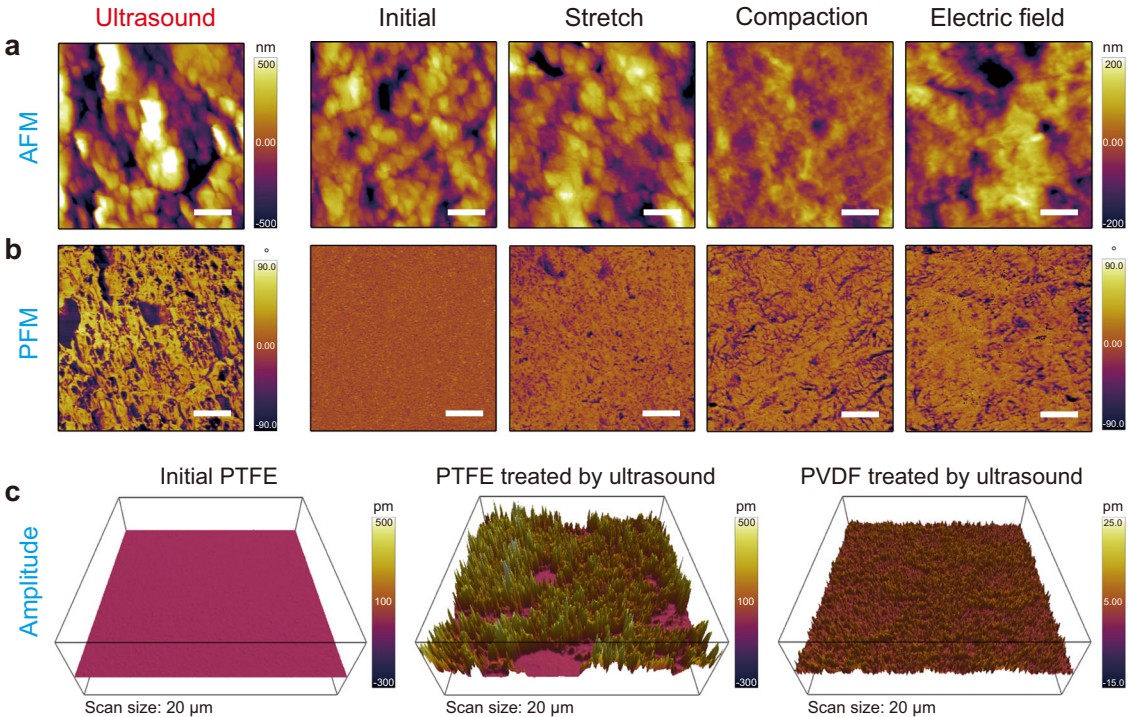

**Fig. 2 Piezoresponse force microscopy of PTFE and PTFE electret. a** Atom force microscopy images (AFM) and **b** PFM phase images of PTFE membrane after different treatments. Ultrasound: A piece of PTFE membrane and 50 mL deionized water were ultrasound irradiated for 1 h. Stretch: A piece of PTFE membrane was stretched by hand. Compaction: A piece of PTFE membrane was compacted for 5 min using a tableting machine with a pressure of 25 Mpa. Electric field: A piece of PTFE membrane was polarized in a parallel electric field (about 134 kV/m) for 2 h. **c** Amplitude mapping of vertical PFM overlaid on 3D topography of PTFE (color bar ranges from −300–500 pm) vs PVDF (color bar ranges from −15–25 pm). Scale bars in **a** and **b** are 4 μm.

conductive AFM cantilever tip, and quality factor, respectively, a higher piezoelectric amplitude, $A$, indicates a larger piezoelectric coefficient, $d_{33}$, which relates the volume change of a piezoelectric material when subject to an electric field. These results demonstrate that the pressure and/or electric fields generated during ultrasonic irradiation transform inert PTFE particles into PTFE piezoelectric electrets with sonocatalytic activity (Figs. 2 and 3).

**Piezoelectric properties of the PTFE and ROS generation**. The piezoelectric properties of the PTFE electrets were characterized by applying an external force and measuring the open circuit-voltage (Fig. 3a, b). Open circuit-voltage ($V_{OC}$) increased monotonically from 0.5 to 1.8 V as the applied force was increased from 0.156 to 0.624 N/cm$^2$. In addition, reproducible voltages (~1 V) were observed when the activated PTFE membrane was ultrasonically irradiated (Fig. 3c). The ultrasonically induced voltage is lower than the actual voltage produced due to aqueous ions or water molecules adsorbing on the electrode surface and forming a screening layer[38]. In summary, sustained ultrasonic pressure waves can continuously stimulate piezoelectric PTFE electrets to produce a rapidly ($f = 40$ kHz; $t = 25$ μs) alternating internal PTFE voltage, which can effectively drive charges to the PTFE-water interface and ultimately generate reactive oxygen species (ROS)[39–41].

To characterize the ROS produced, both ESR and specific oxidant chemical probes were used here. Under air atmosphere, PTFE piezocatalysis yielded ESR quadruplet DMPO-•OH peaks, sextuplet DMPO/DMSO-•O$_2^-$ peaks, and triplet TEMP-$^1$O$_2$ peaks as displayed in Fig. 3d, e, f, respectively. In contrast, these ROS were negligible in absence of PTFE or ultrasound irradiation (Supplementary Fig. 4a-c), and in the case of TiO$_2$ ultrasound irradiation (Supplementary Fig. 4d). Notably, ESR quadruplet DMPO-•OH peaks and quintuplet DMPO-•H peaks could also be

observed under Ar atmosphere indicating O$_2$ is unnecessary for PTFE piezocatalytic ROS production, which is desired in an anaerobic environment (Fig. 3g).

Based on these results, we hypothesized a mechanism for ROS generation via ultrasound-driven piezocatalysis. The permanent polarization of piezoelectric PTFE electrets is accomplished by exposure to high acoustic pressure fields and/or electric fields during ultrasonic irradiation. Meanwhile, the polarization charges in the piezoelectric PTFE electrets are primarily a result of space or surface charges, and the creation of space-charge (or surface charge) electrets is achieved by injecting (or depositing) charge carriers via the high pressures, temperatures, and/or electric fields generated during ultrasound irradiation (Supplementary Fig. 5). Then, when the piezoelectric PTFE is mechanically stimulated rapidly and periodically during ultrasonic irradiation, the polarization magnitude will rapidly oscillate with the dynamic pressure field. Subsequently, the space charges and surface charges will be released as free charges such as separated electron-hole pairs, which may interact with water molecules to produce homogeneous ROS[9,42]. The individual chemical reaction processes are displayed in Eqs. (1–6).

$$PTFE + H_2O \rightarrow poled\ PTFE + \bullet OH + H^+ + e^- \quad (1)$$

$$O_2 + e^- \rightarrow \bullet O_2^- \quad (2)$$

$$H_2O + e^- \rightarrow OH^- + \bullet H \quad (3)$$

$$2\bullet O_2^- + 2H_2O \rightarrow H_2O_2 + 2OH^- + ^1O_2 \quad (4)$$

$$H\bullet + O_2 \rightarrow HO_2 \quad (5)$$

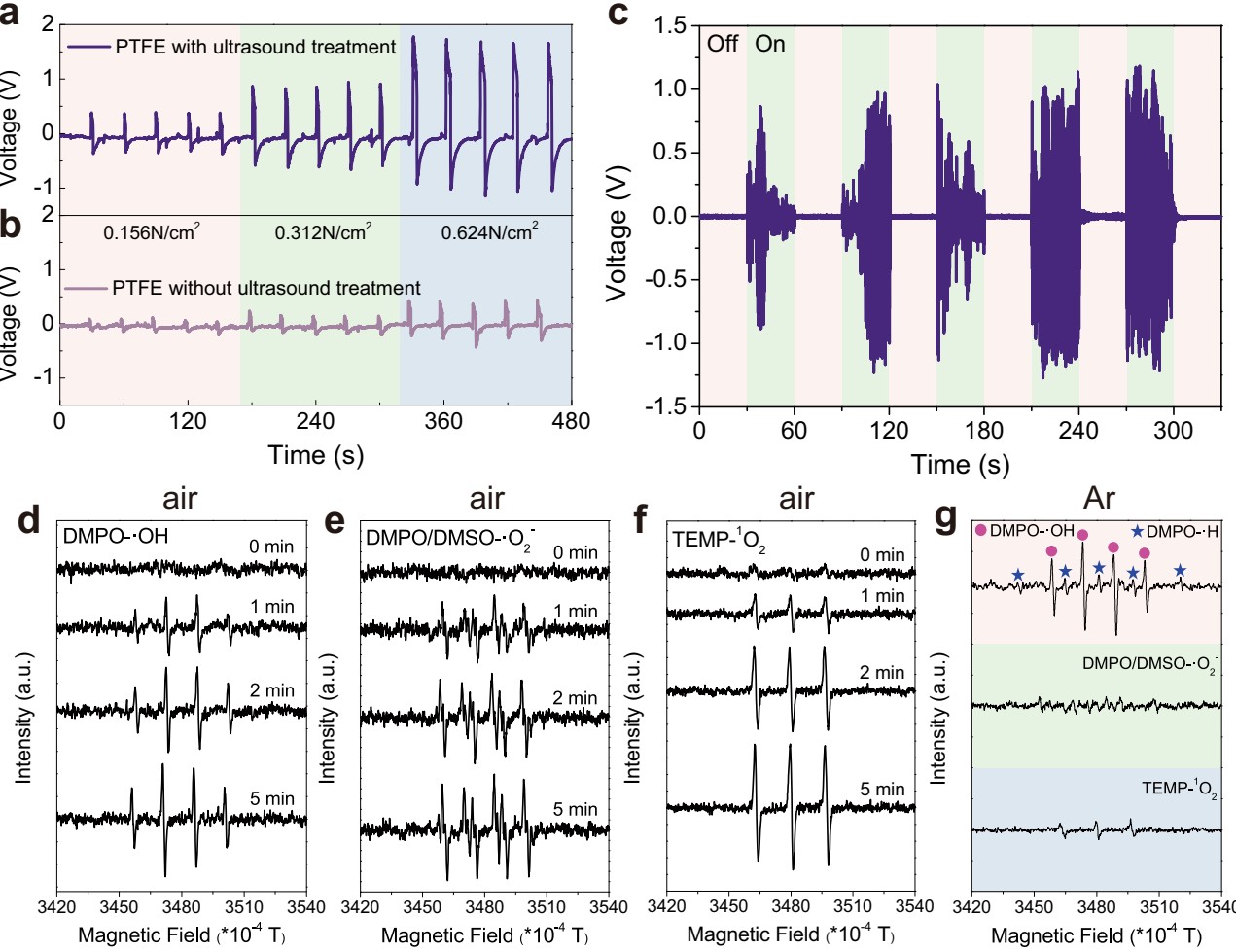

**Fig. 3 Piezoelectric properties of the PTFE and ROS generation. a** Voltage output of the PTFE membrane activated with ultrasound treatment and **b** without ultrasound activation under compression by three different pressure forces. **c** Reproducible voltage output of the PTFE membrane when the ultrasonic cleaner was turned on and off. Schematic illustration of PTFE membrane device see Supplementary Fig. 3. **d**, **e**, **f** Under air and **g** Ar atmosphere, ESR signals for DMPO-•OH, DMPO-•H, DMPO/DMSO-•O$_2$⁻, TEMP-$^1$O$_2$ over PTFE powders under ultrasound irradiation.

$$H\bullet + \bullet HO_2 \rightarrow H_2 + {}^1O_2 \tag{6}$$

**Potential application of piezoelectric PTFE.** Notably, the generated radicals by piezocatalytic PTFE include the •OH, •O$_2$⁻, $^1$O$_2$, and •H, which are among the strongest aqueous redox species that have utility in a range of biological, chemical, and environmental applications. For example, piezocatalysis has been utilized in wastewater purification as mechanical vibration that drives the piezocatalysis is a green energy resource[43–46]. The piezocatalytic decomposition of methyl orange (MO) dye as a function of time during ultrasound irradiation of PTFE electret particles is displayed in Fig. 4a. The piezocatalytic MO removal reaches 89.7 ± 2.9% after 60 min with a pseudo-first-order rate constant of 2.81 h⁻¹ that is >50 times that of ultrasound alone (0.053 h⁻¹), ultrasonicated polyethylene (PE) particles (0.057 h⁻¹) and ultrasonicated TiO$_2$ particles (0.059 h⁻¹), indicating the piezocatalytic effect is specific to PTFE electret particles. PVDF is the classical polymer piezocatalytic material, nonetheless, we found that PVDF particles yielded a piezocatalytic MO degradation rate constant of 0.175 h⁻¹, ~16 times lower than that of PTFE particles, which matched well with the PTFE piezoelectric amplitude (154.7 ± 77.8 pm) being 23.8 times higher than that of the PVDF amplitude (6.5 ± 1.2 pm)

(Fig. 2c). In the absence of ultrasound, the above-mentioned catalysts displayed negligible MO removal, indicating the piezocatalytic reaction required ultrasonic stimulation (Supplementary Fig. 6).

The relationship between ultrasound power and ROS generation, piezocatalytic activity and PTFE PFM was also investigated. The ESR signals intensity for DMPO-•OH were enhanced when the ultrasound power was increased from 0.5 to 2 W/cm² as shown in Supplementary Fig. 7a. Meanwhile, the percentage MO removal increased from 8.3 ± 3.3% to 41.8 ± 1.9% (Supplementary Fig. 7b). According to the equation $q = d_{33}T$, where $q$ and $T$ are piezoelectric charges and the external stress, an increased acoustic amplitude will increase stress $T$, which will induce more transient piezoelectric PTFE surface charges, resulting in a higher MO removal and ROS generation[16]. However, the results of the PTFE PFM showed that increasing ultrasound power could not proportionally improve the localized polarization strength. It is probably that a low ultrasound power could sufficiently activate PTFE, but the piezocatalytic activities of activated PTFE are related to the applied pressure (Supplementary Fig. 7c).

Additionally, acid orange 7 (AO7; anionic dye), methylene blue (MB; cationic dye), Nitrobenzene (NB, persistent and toxic pollutant), and 4-chlorophenol (4-CP) were selected to probe the breadth of PTFE piezocatalytic activity and ROS production.

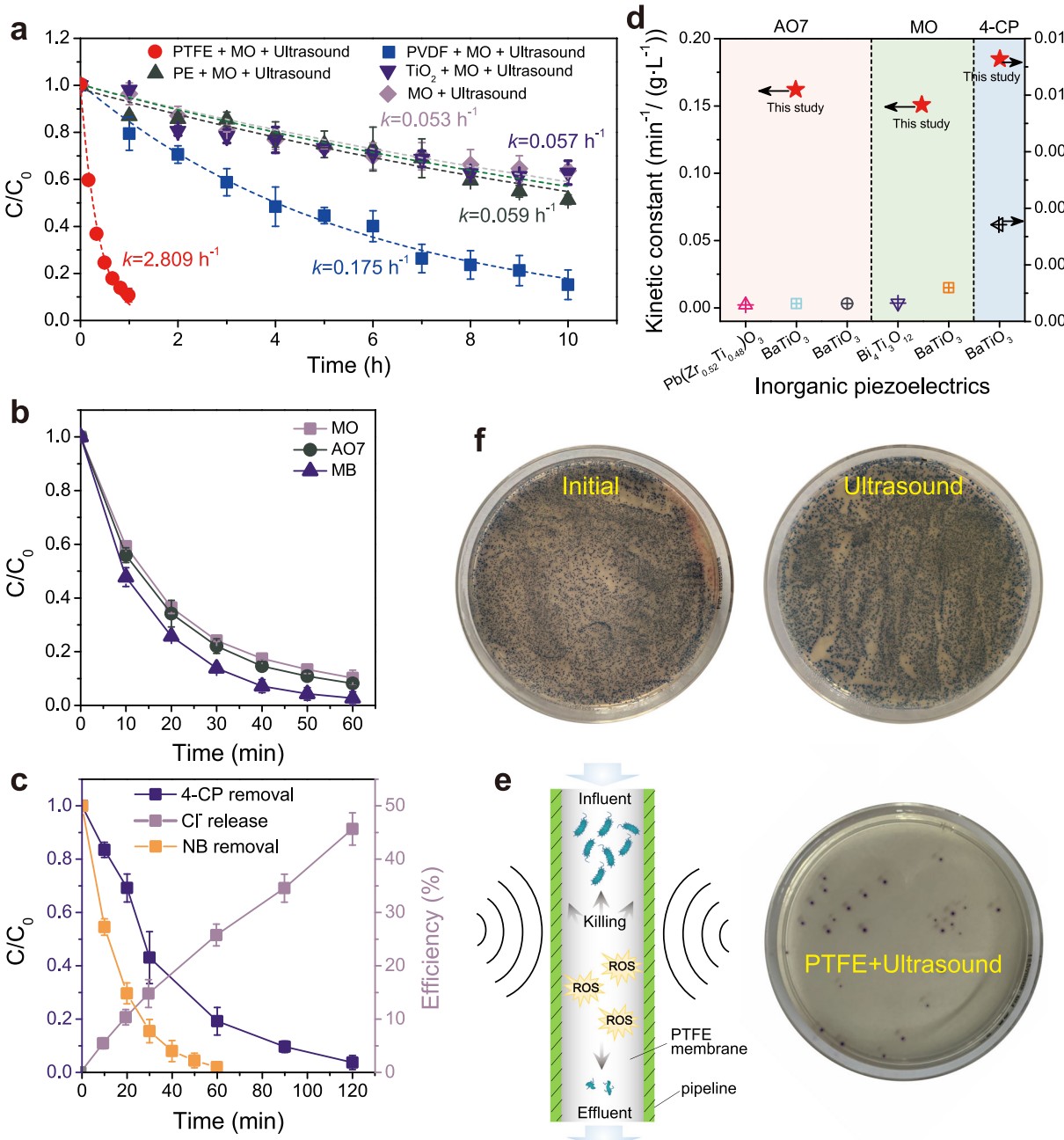

**Fig. 4 Potential application of PTFE piezocatalyst. a** Comparison of catalytic activities between PTFE and PVDF, Polyethylene (PE), TiO₂. Piezocatalytic transformation of **b** dyes and **c** NB, 4-CP. **d** PTFE piezocatalytic activities compared with those of inorganic piezoelectric materials, detail information see Supplementary Table 1. **e** Schematic illustration of PTFE membrane for drinking water disinfection under ultrasound irradiation. **f** Photography of *E. coli* colonies on agar culture plates before and after ultrasound.

Near complete (>90%) transformation was achieved for all compounds (Fig. 4b) with ~50% dechlorination for 4-CP (Fig. 4c) displaying the non-selective nature of the strong ROS produced. Moreover, the PTFE piezocatalytic activity is significantly greater than all other currently reported piezoelectric catalysts under the similar experimental conditions (Fig. 4d, Supplementary Table 1). In summary, PTFE electret particles are effective, chemically stable, and environmentally friendly piezoelectric catalysts that offer many compelling advantages for a number of applications.

ROS are also known to inactivate bacteria, thus many strategies are under development to efficiently produce ROS in situ while avoiding potential negative side-effects such as disinfection by-product formation. For example, drinking water disinfection is

vital for preventing transmission of waterborne diseases[47–49]. In many cases, a residual disinfectant is added to treatment plant effluent to prevent pathogen growth during distribution. However, this may result in disinfectant or disinfection by-product contamination of drinking water. Thus, one solution would be to generate ROS within the water distribution pipeline. Here, we hypothesized that coating the inner water of a water pipe with activated PTFE electret could be used to piezoelectrically generate ROS for drinking water disinfection (Fig. 4e). To simulate this situation, a PTFE membrane was attached to inner wall of the beaker and filled with a solution of *E. coli*. The PTFE coated *E. coli* beaker was then ultrasonically irradiated for 15 min, resulting in inactivation of 99.7% of the initial bacteria (Fig. 4f).

In contrast, the control and ultrasound only systems did not exhibit any obvious antibacterial activity. In addition, SEM images indicated the bacteria cell structure was significantly degraded in the (PTFE + ultrasound) system whereas the cells were predominantly intact in the ultrasound alone system (Supplementary Fig. 8). In real applications, converting PTFE particles into porous filters can broaden the range of potential applications for water purification. The electrospun PTFE filters can simultaneously trap and inactivate the bacteria in situ under ultrasound irradiation (Supplementary Fig. 9). Additionally, we also found PTFE + ultrasound system could inactivate fungi (Candida) and MKN45 cells (human gastric cancer cell) (Supplementary Fig. 8).

Ultrasound has been widely used in biomedical applications such as sonodynamic therapy due to its high tissue penetrability, directed orientation, and ability to trigger in situ ROS generation[10–12]. Generally, molecular sonosensitizers such as hematoporphyrin, Rose Bengal, or B-TiO$_2$-X are needed to enhance ROS yields[50]. Thus similarly, ultrasound-activated PTFE electrets have potential to be utilized as a biocompatible ROS sonosensitizer during sonodynamic therapy. As a proof of concept, we evaluated the PTFE piezocatalytic ROS production using a commercial ultrasound therapy device and porcine epidermis, which is similar to human skin in terms of anatomy and composition (Supplementary Fig. 10a). Through-tissue PTFE particle sonocatalysis displayed high ESR •OH, •O$_2^-$, and $^1$O$_2$ generation as compared to BaTiO$_3$ sonocatalysis and sonolysis alone, which had negligible ROS formation (Supplementary Fig. 10b). ROS can be generated with sonolysis alone, but the high ultrasonic power necessary was damaging to the biological tissue. Notably, ROS could also be generated in an anaerobic environment (Fig. 3g), which is desired during killing tumor cells that exist in anoxic microenvironments. Therefore, PTFE activated by low-power ultrasound can generate strongly oxidizing ROS as nanomedicine with minimal negative side-effects on tissue, which is desired in sonodynamic therapy (Supplementary Fig. 10c).

In summary, we demonstrated a mild and simple method to ultrasonically induce polarization of inert PTFE into piezocatalytic electrets. The piezocatalytic PTFE was observed to generate strong aqueous radicals under mild ultrasonic irradiation, even in an anaerobic environment. The strongly oxidizing ROS produced rapidly degraded organic pollutants and disinfected drinking water. The piezocatalytic PTFE may also have biomedical applications as a proof of the concept suggested that ultrasound with high tissue penetrability and PTFE as a biocompatible sonosensitizer can be utilized to generate in situ ROS, which may have potential for sonodynamic therapy.

Our initial assessment of environmental PTFE piezocatalytic applications presented here may only be the tip of the iceberg. Piezoelectric PTFE particles and films may have potential in fields such as flexible electronics, wearable sensors, acoustic transformers, biocompatible sonosensitizers, pulse imaging, and non-destructive testing[51]. Ultimately, the potential applications will be revealed by future in-depth studies to improve our mechanistic understanding of PTFE particle piezoelectric properties and activity.

## Methods

**PTFE powders characterization.** The morphology of PTFE powders was performed by scanning electron microscopy (Quanta 250 FEG). X-ray photoelectron spectrometry (XPS) was performed on a PHI5000 VersaProbe XPS (UIVAC-PHI, Japan) using a monochromatic Al-K X-ray source. The particle size distribution of the PTFE were estimated using a laser particle size analyzer (Malvern, Nano ZS90).

**PFM characteration of PTFE and PVDF.** PTFE powders are difficult to pretreat with tensile and electric field, and on top of this powders are not suitable for PFM analysis. Thus, a PTFE membrane was used instead of PTFE powder for the PFM

characterization experiments. A piece of 20 mm diameter PTFE membrane was put through a range of treatments including ultrasound irradiation, compaction, tensile testing, and electric field prior to PFM analysis. The PTFE and PVDF membrane piezoelectric properties were obtained using a commercial piezoresponse force microscopy (PFM) (Asylum Research Cypher-ES). PFM is a modification of atomic force microscopy (AFM), with application of a 1.0 V alternating drive voltage on the conductive AFM tip.

**Electrical PTFE characterization.** A piece of 10 mm diameter PTFE membrane was treated with ultrasound irradiation for 1 h, then the upper and lower sides of the PTFE membrane were connected to copper meshes using a conductive carbon epoxy. The copper meshes were connected to a digital multimeter (DMM6500, Keithley). To electrically characterize the PTFE, 50 g (0.156 N/cm$^2$), 100 g (0.312 N/cm$^2$), or 200 g (0.624 N/cm$^2$) weights were pressed onto the copper mesh on the top of the PTFE, then the voltage is recorded by the multimeter. The voltage signals of the PTFE membrane were also measured in the ultrasonic cleaner (Branson 3800-CPXH, 40 kHz, 110 W) in the absence and presence of irradiation.

**Catalytic degradation of dyes by PTFE.** PTFE powders (12.5 mg) (Macklin, 1–5 μm) were placed in a 100 mL beaker and then evenly dispersed with a medicinal spoon. The subsequent experiments involving PTFE were completed in the same manner unless otherwise specified. Next, 50 mL of the 5 mg/L target dyes (e.g., MO, MB, AO7) was poured into the beaker. The inner diameter of the beaker is about 4.8 cm and the height of water in the ultrasonic bath is about 11 cm. The beaker is suspended within the ultrasonic bath such that the water level of the solution in the beaker is the same as the water level in the ultrasonic cleaner (Supplementary Fig. 11). Subsequently, the beaker containing PTFE and dye was irradiated in the ultrasonic cleaner without magnetic stirring. During ultrasonic irradiation, at 10 min intervals, aliquots (1 mL) were sampled and centrifuged to separate the PTFE and obtain a transparent dye solution. The residual dye concentration in the supernatant was analyzed at the maximum absorption wavelength by the UV-vis spectrophotometer.

**Catalytic degradation of 4-CP by PTFE.** To prepare the experiments, 50 mL of 25 mg/L 4-CP was poured into a beaker containing 12.5 mg PTFE powders (Macklin, 1–5 μm). Subsequently, the beaker containing the PTFE and 4-CP was irradiated in the ultrasonic cleaner without magnetic stirring. During ultrasonic irradiation, at 10, 20, 30, 60, 90, and 120 min, aliquots (1 mL) were sampled and centrifuged to separate the PTFE and obtain a transparent 4-CP solution. The residual 4-CP concentration was analyzed by high performance liquid chromatography (HPLC, UltiMate 3000, Thermo Scientific, U.S.A.) equipped with a C18 column (100 × 4.6 mm, 3.5 mm; Agilent, USA). The mobile phase was water: methanol (30:70, v/v) eluted at a flow rate of 0.5 mL min$^{-1}$.

Chloride ion production was monitored by ion chromatography (Dionex 1100) using an IonPac AS11-HC (4 × 250 mm) column with a 20 mM KOH mobile phase at a flow rate of 1.0 mL min$^{-1}$. The dechlorination ratio ($R_{Cl}$) was then calculated as follows:

$$R_{Cl} = [C_{Cl^-}/(C_{04-CP} \times 35.5/128.5)] \times 100\% \qquad (7)$$

where, $C_{Cl^-}$ is the concentration of Cl$^-$ in aqueous solution, and $C_{04-CP}$ is the initial concentration of 4-CP (mg/L). 35.5 and 128.5 are the relative atomic mass of chloride and the relative molecular mass of 4-CP, respectively.

**Catalytic degradation of NB by PTFE.** 12.5 mg PTFE particles were placed in a 100 mL beaker and then evenly dispersed with a medicinal spoon. Next, 50 mL of the 10 mg/L NB solution was poured into the beaker. Subsequently, the beaker containing PTFE and NB was irradiated in the ultrasonic cleaner. During ultrasonic irradiation, at 10 min intervals, aliquots (1 mL) were sampled and centrifuged to separate the PTFE and obtain a NB dye solution. The residual NB concentration was analyzed by high performance liquid chromatography (HPLC, UltiMate 3000, Thermo Scientific, U.S.A.) equipped with a C18 column (100 × 4.6 mm, 3.5 mm; Agilent, USA). The mobile phase was water: methanol (30:70, v/v) eluted at a flow rate of 0.5 mL min$^{-1}$.

**Reactive oxygen species (ROS) analysis in the ultrasonic cleaner.** Reactive oxygen species were detected by the electron spin resonance (ESR) at ambient temperature. Hydroxyl radicals (•OH) and hydrogen atoms (•H) were trapped by DMPO. Singlet oxygen ($^1$O$_2$) was trapped by TEMP. Superoxide radical (•O$_2^-$) was trapped by DMPO/DMSO. In order to avoid the scavenging reaction of •H with oxygen, the deionized water was deoxygenated by bubbling Ar gas for 30 min before ultrasonic irradiation. The reactions were carried out in 1.5 mL centrifuge tubes with 0.5 mL deionized water containing 4 g/L catalyst (i.e., PTFE, TiO$_2$) and 100 mM DMPO were employed to detect DMPO-•OH and DMPO-•H by ESR. Deionized water (0.5 mL) containing 4 g/L catalyst (i.e., PTFE, TiO$_2$) and 50 mM TEMP were employed to detect TEMP-$^1$O$_2$ by ESR. Deionized water (0.5 mL) containing 4 g/L catalyst (i.e., PTFE, TiO$_2$), 100 mM DMPO and 1 M DMSO were employed to detect DMPO/DMSO-•O$_2^-$ by ESR. After 1, 2 and 5 min of ultrasonic

irradiation in the ultrasonic cleaner, the solutions were analyzed by a Bruker EMX-10/12 spectrometer (Germany).

**Disinfection of bacteria experiments**. A piece of PTFE membrane was tightly attached to the inner wall of the 100 mL beaker. 50 mL of *E. coli* suspensions at a concentration of $10^5$ CFU/mL was placed in a 100 mL beaker. Subsequently, the beaker containing PTFE membrane and *E. coli* was irradiated in the ultrasonic cleaner for 15 min. Finally, 100 μL of *E. coli* suspension was added to a standard agar culture medium and incubated at 37 °C for 12 h.

**Reporting summary**. Further information on research design is available in the Nature Research Reporting Summary linked to this article.

## Data availability
The authors declare that the data supporting the findings of this study are available within the paper and its supplementary information files. Any other relevant data are available from the corresponding author upon reasonable request. Source data are provided with this paper.

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

## Acknowledgements

We would like to thank S.T. Zhang, Q.D. Shen, J. Tu, and W. W. Wang for fruitful discussion. We also would like to thank Z.N. Xi and C. Li for PFM measurements. The work was financially supported by National Natural Science Foundation of China (21976085), National Key Research and Development Program of China (Grant No. 2016YFA0203104, 2017YFE010720), Jiangsu Science and Technology Department (Grants: BE2017710).

## Author contributions

Y.F.W. finished experiments and wrote the manuscript and G.D.G. conceived the idea, designed experiments, and revised manuscript. C.D.V. discussed data and revised manuscript. Y.M.X. performed PFM experiments. Y.M.X., S.S.D., P.W., W.C., Z.D.L., D.J.Y., B.C.P., and D.W. contributed to experiments and data analysis. All the authors discussed the results and commented on the manuscript.

## Competing interests

G.D.G. and Y.F.W. are inventors on a patent application (202010014724.7) that covers the use ultrasound waves to drive the PTFE to generate the ROS and electricity.
