## [Peer Review File · Nature Communications]

REVIEWER COMMENTS

Reviewer #1 (Remarks to the Author):

Authors report ultrasonic generation of reactive oxygen species from poly(tetrafluoroethylene) particle and film. They have characterized the formation of hydroxyl radicals, superoxide and singlet oxygen. They have used this approach for environmental pollution control and bacterial therapy. Although the theme of this type piezocatalysis is important and focus of several research groups, here the authors have demonstrated that this approach can be extended to thin film of poly(tetrafluoroethylene) and for bacterial therapy. This work is important and publishable with following suggested modifications:

- 1) Authors should estimate different radicals and compared with some of the well known peizocatalysts. In addition they should estimate different radicals and compared for particle vs film.
- 2) Real application of this approach for water purification requires conversion of poly(tetrafluoroethylene) to porous candle followed by ultrasound-based bacterial disinfection. This should be experimentally demonstrated.

Reviewer #2 (Remarks to the Author):

This work demonstrates that poly(tetrafluoroethylene (PTFE) particles could possesses a piezoelectric property by applied a convenient ultrasound process. The topic is interesting and unique. The authors demonstrated the piezoresponse of PTFE using PFM. Besides, ROS species have been demonstrated by EPR. The PTFE could be used for a potential medical therapy by sonodynamic process. However, some severe fundamental issues did not justify how PTFE exhibits piezoelectric property. The manuscript does not present clearly.

1. If the PTFE has piezoelectric property, the non-centrosymmetric structure should be demonstrated using a molecule structure. Justification should be clearly explained.
2. The sonocatalytic degradation process could participate in the reactions for the generation of ROS (as refereed by: Applied Catalysis B: Environmental Volume 243, 2019, 629-640, Please explain.
3. Authors explain that "...considering the abundance of C-F bonds in PTFE and the strong dipole of the C-F bond, the extremely high pressures and electric fields generated during ultrasonication may quasi-permanently polarize localized regions of the PTFE creating a piezoelectric electret material...." The C-F bond could randomly be distributed. Such local dipoles could be canceled out each other. The theoretical calculation regarding the net dipole should be clearly explained.
4. The PFM response did not present a 180° out-of-phase piezoresponse to the driving voltage. Thus, I doubt that the signal may not be attributed to the piezoelectric response.
5. If a parallel electric field can polarize the PTFE, its P-E curve should be presented to identify the ferroelectric properties. Does PTFE belong to piezoelectric material or ferroelectric?
6. After applied the compressive and tensile stress to the PTFE, the deformation could create structural defects on the PTFE's surface. As a consequence, the mechanical force could induce the surface charges on the PTFE. Notably, PTFE has fluoride atoms with high electronegativity that it tends to hold the

electrons in the carbon-fluorine bonds closely. Therefore, the PFM measurement could difficult to distinguish the signal is attributed to the piezoresponse.

7. The PTFE should be proceeded with the aging test after applied the mechanical strain and measured their PE curve and PFM for comparison. The surface charges could strongly be related to aging time.

8. The different pressures and electric fields should be evaluated during the ultrasonic process to compare the following parameters: ROS, degradation performance, PE curve, and piezoresponse.

Reviewer #3 (Remarks to the Author):

This paper presents interesting results on ROS generation using ultra sound activation of piezo electric materials.

The authors might include some remarks on how the ROS might be distributed into the bulk solution, considering that with a PTFE membrane the ROS is produced on its surface. This is pertinent to how this might be used in, for instance, biomedical therapy; would that involve injection of PTFE particles into the site targeted for therapy??

Using ultrasound to induce piezo-electric properties in the polymer materials, presumable leads to randomly oriented "poling" of the materials, unlike traditional poling using electric field applied at elevated temperatures. A comment related to that would be good to include. The random orientation of electrets will not, of course, affect ROS production using subsequent exposure to ultra sound.

The results presented, for instance in Figure 4. includes a control where only ultrasound was used but no piezo materials. That is good, but it also requires controls WITH those materials present but without ultra sound.

The results shown in figure 4 for the bacteria are not clear - the photographs of the agar plates are not of high quality.

Methods section: I found a lack of detail such as the geometry of the samples/containers. For instance in the bacteria experiments, the generation of ROS would have been at the membrane surface lining the beaker. How large was this beaker in diameter. We are told it was a 100 ml beaker but this provides little insight into the distance the ROS has to diffuse to react with the bacteria in suspension.

Overall, the results presented would be of high interest to many researchers interested in water remediation and biomedical treatment. There would appear to be many exciting applications, including pretreatment of feed water in membrane based water treatment plants.

Response to Reviewers' Comments

Reviewer #1 (Remarks to the Author)

Authors report ultrasonic generation of reactive oxygen species from poly(tetrafluoroethylene) particle and film. They have characterized the formation of hydroxyl radicals, superoxide and singlet oxygen. They have used this approach for environmental pollution control and bacterial therapy. Although the theme of this type piezocatalysis is important and focus of several research groups, here the authors have demonstrated that this approach can be extended to thin film of poly(tetrafluoroethylene) and for bacterial therapy. **This work is important and publishable** with following suggested modifications.

Response:

We greatly appreciate Reviewer 1's positive evaluation and kind recommendation in regard to publishing our manuscript in *Nature communications*. We have addressed all of Reviewer 1's comments and concerns in detail, and the corresponding revisions have been made in the revised Manuscript and Supplementary Information. We hope that your concerns have been eased after our detailed explanations and revisions.

General issues:

Comment 1.1: Authors should estimate different radicals and compared with some of the well known piezocatalysts. In addition they should estimate different radicals and compared for particle vs film.

Response 1.1: We thank the reviewer for this comment. In the PTFE system, hydroxyl radicals ($\bullet\text{OH}$), superoxide ($\bullet\text{O}_2^-$) and singlet oxygen ($^1\text{O}_2$) were the main reactive oxygen species (ROS). As compared with other well-known piezocatalysts from previous research studies, we find there is no difference in the type of radical species generated between PTFE and other piezocatalysts (*Nat. Commun.* 2020, 11, 1328; *Environ. Sci. Technol.* 2017, 51, 6560-6569; *Chemosphere* 2018, 193, 1143-1148; *Appl. Catal. B-Environ.* 2017, 219, 550-562; *Appl. Catal. B-Environ.* 2020, 279, 119353). The formation of ROS is a universal reaction during piezocatalysis. The previous studies did not directly quantify the ROS concentration, thus it is difficult to compare the concentration between PTFE and other inorganic piezocatalysts. However, we did compare the piezocatalytic degradation rate of various aqueous pollutants using different piezocatalysts (see Table S1).

Obviously, the results indicate that PTFE exhibited a superior piezocatalytic activity than other piezocatalysts (i.e. significantly greater catalyst normalized degradation rate constants) in regard to organic degradation and thus ROS production.

Table S1. Summary of piezocatalyst kinetic rate constants for organic degradation

Piezocatalyst	Conditions	Pseudo-first order rate constant (min ⁻¹)	Catalyst normalized rate constant min ⁻¹ /(g·L ⁻¹)	Ref.
Pb(Zr _{0.52} Ti _{0.48})O ₃	[catalyst] ₀ = 12.5 g/L [AO7] ₀ = 30 μM (10.5 mg/L)	0.0279	0.0022	1
Bi ₄ Ti ₃ O ₁₂	[catalyst] ₀ = 1.33 g/L [MO] ₀ = 10 μM (3.27 mg/L)	4.6×10 ⁻³	0.0035	2
BaTiO ₃	[catalyst] ₀ = 1 g/L [MO] ₀ = 5 mg/L	0.015	0.0150	3
BaTiO ₃	[catalyst] ₀ = 10 g/L [AO7] ₀ = 57 μM (20 mg/L)	0.0313	0.0031	4
BaTiO ₃	[catalyst] ₀ = 9 g/L [AO7] ₀ = 57 μM (20 mg/L)	0.0285	0.0032	5
BaTiO ₃	[catalyst] ₀ = 2 g/L [4-CP] ₀ = 25 mg/L	0.0101	0.0051	6
PTFE	[catalyst] ₀ = 0.25 g/L [AO7] ₀ = 30 μM (10.5 mg/L)	0.0403	0.1612	This study
PTFE	[catalyst] ₀ = 0.25 g/L [MO] ₀ = 5 mg/L	0.0377	0.1508	This study
PTFE	[catalyst] ₀ = 2 g/L [4-CP] ₀ = 25 mg/L	0.0277	0.0139	This study

In addition, we also investigated the ROS generation by piezocatalytic PTFE membrane (**Fig. R1**). The results showed that •OH, •O₂⁻ and ¹O₂ were also detected in the PTFE membrane system, which indicated that there was no obvious difference in ROS species generated by PTFE particles and membranes.

Fig. R1 ESR signals for DMPO-•OH, DMPO-•H, DMPO/DMSO-•O₂⁻, TEMP-¹O₂ in the solution over the PTFE membrane under ultrasound irradiation.

Comment 1.2: Real application of this approach for water purification requires conversion of poly(tetrafluoroethylene) to porous candle followed by ultrasound-based bacterial disinfection. This should be experimentally demonstrated.

Response 1.2: We thank the reviewer for the excellent suggestion. Indeed, recovering catalyst particles during water purification is an expected issue and transforming PTFE particles into macroscopic porous structures will widen its application potential for water purification and other processes. Here, we used an electrospinning technique to convert PTFE particles into a PTFE fiber membrane. According to the reviewer's suggestion, we investigated the disinfection of bacteria using the piezocatalytic PTFE fiber filter.

Action: The results of the piezocatalytic PTFE filter bacterial disinfection experiments are displayed in **Fig. R2**. The following sentences were added into the revised manuscript on **Page 13**. The corresponding electrospinning method to prepare the PTFE fiber filters has also been added into the Supplementary Information on **Page 2**.

In real applications, converting PTFE particles into porous filters can broaden the range of potential applications for water purification. The electrospun PTFE filters can simultaneously trap and inactivate the bacteria in situ under ultrasound irradiation.

Fig. R2 (a, b) SEM image of PTFE fiber membranes prepared by electrospinning and the corresponding EDS mappings of (c) C and (d) F of the region enclosed by the yellow square. (e) The energy dispersive spectrum (EDS) of the yellow square region. (f) ATR FTIR of PTFE fiber membranes. (g) Photography of *E.coli* colonies on agar culture plates before and after ultrasound with PTFE fiber membranes. The results showed that PTFE fiber membranes could achieve inactivation of about 85.1% of the initial *E.coli*. (This figure has been added into the Supplementary Information as new Supplementary Fig. 8.)

Reviewer #2 (Remarks to the Author)

This work demonstrates that poly(tetrafluoroethylene (PTFE) particles could possess a piezoelectric property by applying a convenient ultrasound process. **The topic is interesting and unique.** The authors demonstrated the piezoresponse of PTFE using PFM. Besides, ROS species have been demonstrated by EPR. The PTFE could be used for a potential medical therapy by sonodynamic process. However, some severe fundamental issues did not justify how PTFE exhibits piezoelectric property. The manuscript does not present clearly.

Response:

We thank reviewer 2 for the positive evaluation of our manuscript. We have revised our manuscript according to your comments and the detailed responses are given in the comments 1 to 8.

General issues:

Comment 2.1: If the PTFE has piezoelectric property, the non-centrosymmetric structure should be demonstrated using a molecule structure. Justification should be clearly explained.

Response 2.1: We thank the reviewer for this constructive comment. The piezoelectric mechanism between PTFE electret and classical piezoelectric materials such as ZnO and BaTiO₃ is different. The piezoelectricity of ZnO originates from its non-centrosymmetric nature (i.e. the ZnO crystals do not have a center of symmetry in their structure), which results in permanent electric dipoles within the material (*Appl. Catal. B: Environ.* 2019, 241, 256-269; *Nano Today* 2010, 5, 540-552).

The polarization formation mechanism of polar electrets is similar to traditional piezoelectric materials. Generally, polarization of polar electrets is completed by placement of a material within an electric field at room temperature or a decreasing temperature over an appropriate range. In contrast, polarization of non-polar electrets, such as PTFE, is primarily a result of space-charges, and the charging of space-charge (or surface charge) electrets is usually achieved by injecting (or depositing) charge carriers via corona discharge, electrical breakdown radiation, light, pressure, or heat (*Electrets (Topics in Applied Physics)* (Springer-Verlag, Berlin, Heidelberg, New York, 1980)). In our study, ultrasonic irradiation of aqueous solutions generates at least three elements mentioned above, which individually or in combination may be the driving force behind the polarization of the non-polar PTFE electret. Generally, **as the reviewer said in the comment 2.6**, the ultrasonic PTFE

structural deformation could create surface defects, which results in non-polar materials having permanent charges and apparent piezoelectricity. For example, the piezoelectricity of the PTFE electret could be derived from external charge injection or introduction of structural defects rather than its specific molecule structure. In summary, the polarization mechanism of traditional piezoelectric materials and non-polar electrets differs greatly (**Fig. R3**), and the latter tends to have a higher piezoelectric performance.

In our study, we demonstrated a method to induce initially inert PTFE into piezoelectric electrets with a mild and convenient ultrasound process. The piezoresponse of PTFE electret was characterized by PFM. Notably, the high pressure and electric fields generated by ultrasound will create PTFE structural defects (*Adv. Mater.* 2020, 32, 2000006). The electret charge may consist of "real" charges, such as surface-charge layers or space charges; or it may be a "true" macroscopic polarization; or most likely it may be a combination of these two polarization processes. Specifically, when subject to an ultrasonic field, the large and transient compressive and tensile stresses on the PTFE will result in massive structural deformation that will likely create defects or even a microporous structure. Subsequently, the charges and electric fields generated during acoustic cavitation could be injected into these structural/surface defects to ultimately yield a piezoelectrically active PTFE electret (**Fig. R3**).

Fig. R3 Atomic model of the wurtzite-structured ZnO (a) and schematic cross section of electret having deposited surface charges, injected space charges, aligned dipolar charges (or microscopically displaced charges), and compensation charges (b). (*Nano Today* 2010, 5, 540-552; *Jpn. J. Appl. Phys.* 2009, 48, 09KE05; *Electrets (Topics in Applied Physics)* (Springer-Verlag, Berlin, Heidelberg. New York, 1980, 33, pp. 1-11)

Although the initial PTFE as a whole is centrosymmetric, localized regions near the chemical or physical defects may cause PTFE to be in a non-centrosymmetric phase similar to conventional piezoelectric polymers (e.g., polyvinylidene fluoride) (*Cryst. Res. Technol.* 1991, 26(6), 767-781). However, we did not observe release of fluoride ions to the solution within the detection range after the PTFE was treated by ultrasound. Since the average PTFE particle size was 5 μm , it is difficult to investigate the structural changes on the atomic scale. Overall, as opposed to the natural non-centrosymmetric nature of ZnO, the piezoelectricity behavior in the PTFE electret is from deformation nonlinearity and the presence of the defect-related charges (*Soft Matter* 2019, 15, 262; *Jpn. J. Appl. Phys.* 1974, 13, 197).

Action: In the revised version of the manuscript, we added discussion on the mechanism of PTFE exhibiting piezoelectric property under ultrasound irradiation in more detail on **Page 6** as shown in following, and cited the relevant references in their proper places.

This indicates the emergence of PTFE electret domains with strong localized piezoelectric properties. Ultrasonic pressure waves force the formation and quasi-adiabatic collapse of vapor bubbles formed from pre-existing gas nuclei. The transient collapse of acoustic cavitation bubbles can generate extremely high pressures (~100 Mpa) and electric fields (~100 kV/m)^{32,33}. These transient and extreme ultrasonic cavitation pressure waves can cause massive PTFE deformation resulting in permanent structural defects. As a consequence, the concurrent electric fields can generate charges that can be trapped in the PTFE structural defects creating an electret state²⁵. In addition, although the initial PTFE as a whole is centrosymmetric, some of the local regions near these chemical or physical defects may cause PTFE to be in a localized non-centrosymmetric phase³⁴. In agreement with experiments, recent theoretical modeling has reported the piezoelectric behavior of the PTFE electret to be due to the presence of charges, the interaction of Maxwell stress, and deformation nonlinearity^{35,36}.

Comment 2.2: The sonocatalytic degradation process could participate in the reactions for the generation of ROS (as refereed by: Applied Catalysis B: Environmental Volume 243, 2019, 629-640, Please explain.

Response 2.2: We appreciate the reviewer's thoughtful question.

The article mentioned by the reviewer studied the sonocatalytic degradation of Rhodamine B dye by micro- and nano-particles of ZnO. The results also demonstrated the generation of ROS under ultrasonic irradiation of ZnO. However, ZnO is also a classical piezoelectric material. Many recent research studies have focused on the piezocatalytic degradation activity of ZnO (*Chemosphere* 2018, 193, 1143-1148; *J. Colloid Interf. Sci.* 2020, 577, 290-299.) Therefore, we doubt that the authors have ignored the piezocatalytic activity of ZnO while studying the sonocatalytic degradation process.

In order to eliminate the contribution of sonocatalysis in piezocatalytic process, the ideal sonocatalyst control should have the same chemical composition as the piezocatalyst. For example, tetragonal BaTiO₃ has piezocatalytic activity while cubic BaTiO₃ does not possess piezoelectric activity. Shen et al. reported that the sonochemical degradation of 4-CP by cubic BaTiO₃ was only 16.4% after 120 min, which was comparable to ultrasound alone (*Environ. Sci. Technol.* 2017, 51, 6560-6569). In contrast, tetragonal BaTiO₃ mediated piezocatalysis resulted in 71.1% degradation of 4-CP. These results demonstrate that the sonocatalytic degradation is still low when in the presence of similar non-piezocatalytic particles.

In our study, we use polyethylene (PE) particles, which are similar in structure and reactivity to PTFE as a control to exclude the effect of sonocatalysis. We found that the piezocatalytic MO degradation rate constant of PTFE is >50 times that of PE, indicating the piezocatalytic effect is specific to the PTFE electret.

Comment 2.3: Authors explain that "...considering the abundance of C-F bonds in PTFE and the strong dipole of the C-F bond, the extremely high pressures and electric fields generated during ultrasonication may quasi-permanently polarize localized regions of the PTFE creating a piezoelectric electret material...." The C-F bond could randomly be distributed. Such local dipoles could be canceled out each other. The theoretical calculation regarding the net dipole should be clearly explained.

Response 2.3: We agree with the reviewer that the initial PTFE as a whole is centrosymmetric because dipoles can be canceled out each other. However, the high pressures, temperatures, and electric fields generated by ultrasonic cavitation can cause both large PTFE structural deformation

to create permanent defects and inject charges or form dipoles within PTFE (an effect that would not be expected for classical inorganic piezocatalysts due to their rigid structure). We agree that in the original manuscript we did not explain this mechanism in enough detail.

Please refer to response 2.1 to comment 2.1 for a more detailed explain.

Action: To avoid misleading the reader, we have deleted the corresponding sentences and relevant references ("...considering the abundance of C-F bonds in PTFE and the strong dipole of the C-F bond, the extremely high pressures and electric fields generated during ultrasonication may quasi-permanently polarize localized regions of the PTFE creating a piezoelectric electret material..."). These sentences were replaced with the following explanation to assist the reader in better understanding the formation and piezoelectric properties of the PTFE electret.

These transient and extreme ultrasonic cavitation pressure waves can cause massive PTFE deformation resulting in permanent structural defects. As a consequence, the concurrent electric fields can generate charges that can be trapped in the PTFE structural defects creating an electret state²⁵. In addition, although the initial PTFE as a whole is centrosymmetric, some of the local regions near these chemical or physical defects may cause PTFE to be in a localized non-centrosymmetric phase³⁴. In agreement with experiments, recent theoretical modeling has reported the piezoelectric behavior of the PTFE electret to be due to the presence of charges, the interaction of Maxwell stress, and deformation nonlinearity^{35,36}.

Comment 2.4: The PFM response did not present a 180° out-of-phase piezoresponse to the driving voltage. Thus, I doubt that the signal may not be attributed to the piezoelectric response.

Response 2.4: We thank the reviewer for raising this question. It has been reported that the PFM response of the polypropylene electret and ultrathin MoO₂ electret have ~70° and ~56° phase change, respectively. (*J. Appl. Phys.* 2014, 116, 066820; *Adv. Mater.* 2020, 32, 2000006). PTFE and other electrets may not have presented a 180° out-of-phase piezoresponse due to the following reasons: 1) The surface height can have influence on phase contrast, 2) The signal value of the measured phase contrast is a spatial average, and 3) The ultrasonic driving voltage is not large enough to achieve 180° out-of-phase inversion.

Comment 2.5: If a parallel electric field can polarize the PTFE, its P-E curve should be presented to identify the ferroelectric properties. Does PTFE belong to piezoelectric material or ferroelectric?

Response 2.5: Many thanks for this comments. All insulating materials exhibit electrostriction and the so-called Maxwell stress effect whereby the application of electric field can deform the material (*Phys. Rev. E* 2014, 90, 12603), which explains why an electric field can polarize the PTFE. As the reviewer suggested, the P-E curve was measured to characterize the ferroelectric properties, if any, of the PTFE electrets produced in this study. As shown in **Fig. R4**, the hysteresis loop displays no saturation, thus we cannot tell whether there exist polarization switching in PTFE films because some lossy dielectric materials will display hysteresis loops similar to **Fig. R4**. The lack of spontaneous polarization at zero potential suggests the material is not ferroelectric.

In our study, the transient collapse of acoustic cavitation bubbles during ultrasound irradiation can generate extremely high pressures. After applied the high pressure to the PTFE, the deformation could create structural defects on the PTFE's surface. Subsequently, the defects can attract trapping charges, resulting in apparent piezoelectricity. However, the trapped-charge polarization in PTFE electrets is metastable, while the polarization in ferroelectrics is thermodynamically stable. Therefore, PTFE electret does not belong to ferroelectric class of materials, but belongs to piezoelectric class of materials.

Fig. R4 P-E hysteresis loops of PTFE film before and after ultrasound irradiation.

Comment 2.6: After applied the compressive and tensile stress to the PTFE, the deformation could create structural defects on the PTFE's surface. As a consequence, the mechanical force could induce the surface charges on the PTFE. Notably, PTFE has fluoride atoms with high

electronegativity that it tends to hold the electrons in the carbon-fluorine bonds closely. Therefore, the PFM measurement could difficult to distinguish the signal is attributed to the piezoresponse.

Response 2.6: We thank the reviewer for this comment and fully agree with the reviewer that the surface charges may exist on the PTFE due to mechanical forces. Actually, for ferroelectrics, it is also difficult to avoid shielding surface charges (positive and negative charges) due to the attraction of spontaneous polarized dipoles. It is well known that the piezoresponse of ferroelectrics can be characterized by PFM. Therefore, shielding charges should have negligible influence on the PFM measurement.

As the reviewer stated, the deformation could create both bulk and surface PTFE structural defects after ultrasonic irradiation and the corresponding compressive and tensile stresses it generates. If the PTFE defects are non-centrosymmetric, then they would result in apparent piezoelectricity, which would be observed by PFM. To ensure negligible surface charge, the PTFE is treated in the ultrasonic bath before PFM measurement. Any trapped PTFE surface charges would then react with water or oxygen to generate ROS, resulting in minimal metastable PTFE surface charges during the PFM measurement. Thus, the PFM is solely measuring the PTFE electret properties and not any transient surface charges generated by the activation process.

Comment 2.7: The PTFE should be proceeded with the aging test after applied the mechanical strain and measured their PE curve and PFM for comparison. The surface charges could strongly be related to aging time.

Response 2.7: We thank the reviewer for the helpful comment and fully agree with the reviewer that the surface charges could strongly be related to aging time. Generally, surface charges will reduce with increasing aging time due to the segmental depolarization of orientated dipoles caused by molecular thermodynamic movement (*ACS Appl. Mater. Interfaces* 2016, 8, 23985-23994). The polarization can also be screened over time by ion or molecule adsorption on the surface from air forming a Stern layer (*Chemistry of Materials* 2013, 25, (21), 4215-4223).

As the reviewer suggested, the PTFE aging behavior was investigated using PE curve and PFM measurements as a function of time. The result of PE loop indicates that there is no obvious change in the PTFE properties after 10 days (**Fig. R5a**). As the PTFE treated by ultrasound is electret material, the PE curve is similar to a dielectric material and shows no saturation typical of a

ferroelectric material. In addition, since the PTFE is irradiated by ultrasound in aqueous solution before characterization, the metastable trapped charges and dipoles are reduced because they can react with water or oxygen to generate ROS. Therefore, the PTFE PE curve displays the minimal residual polarization at the macroscopic level. Compared to **Fig. 2c**, the PFM results display that the localized PTFE electret polarization magnitude has decreased after 10 days (**Fig. R5b**), and the reduction of localized microscopic surface charges or polarized dipoles with time indicates they are metastable in nature (**Fig. R5c**).

Fig. R5 *P-E* hysteresis loops (a) and PFM (b) of polarized PTFE film after 10 days

Comment 2.8: The different pressures and electric fields should be evaluated during the ultrasonic process to compare the following parameters: ROS, degradation performance, PE curve, and piezoresponse.

Response 2.8: Thank you for the insightful comment. Actually, it is quite difficult to precisely adjust the pressure and electric field change during ultrasonic irradiation generated acoustic cavitation since the process is so transient (microseconds) and extreme (100s MPa, 100s kV/m, 1000s K, etc.). On the other hand, the ultrasound power can be easily adjusted to control the acoustic pressure field. The magnitude of the acoustic pressure waves will increase with increasing ultrasonic power and if a PTFE particle is interacting directly with a stable cavitation bubble, then as power is increased and the bubble increases in diameter, the PTFE particle may undergo a larger tensile and compressive interactions. At the suggestion of the reviewer, here we further investigated the influence of ultrasound power on ROS, organic degradation performance, PE curve, and PFM.

The ESR signals intensity for DMPO-•OH were enhanced when the ultrasound power increased from 0.5 to 2 W/cm² as shown in **Fig. R6a** below. Accordingly, the MO removal percentage increased from 8.3 ± 3.3% to 41.8 ± 1.9% (**Fig. R6b**). According to the equation $q = d_{33}T$, where q and T are piezoelectric charges and the external stress, an increased acoustic amplitude

will increase stress T , which will induce more piezoelectric PTFE surface charges, resulting in a higher ROS generation and MO removal. However, the results of the PTFE PFM showed that increasing ultrasound power could not proportionally improve the localized polarization strength (Fig. R6c). It is probably that a low ultrasound power could sufficiently activate PTFE, but the piezocatalytic activities of activated PTFE are related to the applied pressure. In addition, the PE curve shows no obvious distinctions under different ultrasound powers, which was expected since the PTFE electret is not a ferroelectric material and the PE curve shows no residual polarization charges (Fig. R7).

Fig. R6 (a) Effect of ultrasound power on $\bullet\text{OH}$ generation, **(b)** piezocatalytic degradation of MO ($[\text{PTFE}]_0 = 0.5 \text{ g/L}$, $[\text{MO}]_0 = 5 \text{ mg/L}$) and **(c)** PFM driven with an ultrasonic therapy device (1.0 MHz, 20% duty cycle, 5 min). (This figure has been added into the Supplementary Information as new Supplementary Fig. 6)

Fig. R7 Effect of ultrasound power on PE curve of PTFE

Action: In the revised version of manuscript, we discuss the effect of ultrasound power on $\bullet\text{OH}$ generation, piezocatalytic degradation of MO, and PFM. The following sentences were added into the revised manuscript on **Page 11-12**.

*The relationship between ultrasound power and ROS generation, piezocatalytic activity and PTFE PFM was also investigated. The ESR signals intensity for DMPO- $\bullet\text{OH}$ were enhanced when the ultrasound power was increased from 0.5 to 2 W/cm² as shown in **Supplementary Fig. 6a**. Meanwhile, the percentage MO removal increased from $8.3 \pm 3.3\%$ to $41.8 \pm 1.9\%$ (**Supplementary Fig. 6b**). According to the equation $q = d_{33}T$, where q and T are piezoelectric charges and the external stress, an increased acoustic amplitude will increase stress T , which will induce more transient piezoelectric PTFE surface charges, resulting in a higher MO removal and ROS generation¹⁶. However, the results of the PTFE PFM showed that increasing ultrasound power could not proportionally improve the localized polarization strength. It is probably that a low ultrasound power could sufficiently activate PTFE, but the piezocatalytic activities of activated PTFE are related to the applied pressure (**Supplementary Fig. 6c**).*

Reviewer #3 (Remarks to the Author)

This paper presents interesting results on ROS generation using ultrasound activation of piezo electric materials.

Response: We are grateful for this kind assessment.

Comment 3.1: The authors might include some remarks on how the ROS might be distributed into the bulk solution, considering that with a PTFE membrane the ROS is produced on its surface. This is pertinent to how this might be used in, for instance, biomedical therapy; would that involve injection of PTFE particles into the site targeted for therapy??

Response: Thanks for the valuable comment. As the reviewer correctly stated, the ROS are generated at the membrane surface and since ROS typically have an ultrashort lifetime (10^{-6} ~ 10^{-9} s) and diffusion distance (1~30 nm) in water (*Angew. Chem. Int. Ed.* 2019, 58(24), 8134-8138; *Trends Cell Biol.* 2008, 18(9), 443-450), they may only be present in the near PTFE surface environment. Therefore, ROS cannot effectively diffuse to the target bacteria or pollutant in solution. However, ultrasound drives rapid solution mixing, which can promote diffusion of bacterial and pollutant to the membrane surface to react with ROS or contact between homogenous ROS and the target species.

For real application *in vivo*, the PTFE nanoparticles should be considered as a nanocatalytic medicine that is injected intravenously into organisms for sonodynamic therapy. Nanoparticles, through the enhanced permeability and retention effect, preferentially accumulate in disease sites (*Annu. Rev. Med.* 2012, 63, 185-198). Then, the PTFE nanoparticles will catalyze ROS generation *in situ* for tumor eradication under the ultrasonic irradiation. Direct intratumoral injection of PTFE nanoparticles is also an effective way to target cancerous tissue and has the advantage of increasing local therapeutic concentration (*Future Med. Chem.* 2015, 7(12), 1503-1510). In addition to cancer therapy, sonodynamic therapy can also be applied to *in vitro* therapies, such as antibacterial therapy and anti-inflammatory therapy (*Adv. Mater.* 2019, 1901778). In this case, an activated PTFE membrane can be directly attached to the target tissue and initiate oxidative damage of pathogenic bacteria by ultrasonic irradiation.

Comment 3.2: Using ultrasound to induce piezo-electric properties in the polymer materials, presumably leads to randomly oriented "poling" of the materials, unlike traditional poling using electric field applied at elevated temperatures. A comment related to that would be good to include. The random orientation of electrets will not, of course, affect ROS production using subsequent exposure to ultrasound.

Response: We thank the reviewer for this comment. In this manuscript, we demonstrated that ultrasound induced inert PTFE transformation into a piezoelectric electret. Meanwhile, we fully agree with the reviewer that piezoelectric PTFE electret will exhibit randomly oriented polarization after ultrasound irradiation. The PFM characterization in Fig. 2 confirms this interpretation. It is worth noting that the piezoelectric PTFE electret charges may consist of surface charges, and space charges. Although these charges are randomly oriented macroscopically, they can have a localized orientation on the microscopic scale (see PFM) and will inject charges into the local aqueous environment under ultrasonic irradiation. Therefore, as stated by the reviewer, the macroscopic random orientation of PTFE electrets will not affect ROS production upon exposure to ultrasound.

The piezoelectric mechanism between PTFE electret and classical piezoelectric materials such as ZnO and BaTiO₃ is different. The piezoelectricity of ZnO originates from its non-centrosymmetric nature (i.e. the ZnO crystals do not have a center of symmetry in their structure), which results in permanent electric dipoles within the material (*Appl. Catal. B: Environ.* 2019, 241, 256-269; *Nano Today* 2010, 5, 540-552). The polarization formation mechanism of polar electrets is similar to traditional piezoelectric materials. Generally, polarization of polar electrets is completed by placement of a material within an electric field at room temperature or a decreasing temperature over an appropriate range. In contrast, polarization of non-polar electrets, such as PTFE, is primarily a result of space-charges, and the charging of space-charge (or surface charge) electrets is usually achieved by injecting (or depositing) charge carriers via corona discharge, electrical breakdown radiation, light, pressure, or heat (*Electrets (Topics in Applied Physics)* (Springer-Verlag, Berlin, Heidelberg. New York, 1980)). In our study, ultrasonic irradiation of aqueous solutions generates at least three elements mentioned above, which individually or in combination may be the driving force behind the polarization of the non-polar PTFE electret. Generally, the ultrasonic PTFE structural deformation could create surface defects, which results in non-polar materials having permanent charges and apparent piezoelectricity. For example, the

piezoelectricity of the PTFE electret could be derived from external charge injection or introduction of structural defects rather than its specific molecule structure. In summary, the polarization mechanism of traditional piezoelectric materials and non-polar electrets differs greatly, and the latter tends to have a higher piezoelectric performance.

Action: In the revised version of the manuscript, we added discussion on the mechanism of PTFE exhibiting piezoelectric property under ultrasound irradiation in more detail on **Page 6** as shown in following, and cited the relevant references in their proper places.

These transient and extreme ultrasonic cavitation pressure waves can cause massive PTFE deformation resulting in permanent structural defects. As a consequence, the concurrent electric fields can generate charges that can be trapped in the PTFE structural defects creating an electret state²⁵. In addition, although the initial PTFE as a whole is centrosymmetric, some of the local regions near these chemical or physical defects may cause PTFE to be in a localized non-centrosymmetric phase³⁴. In agreement with experiments, recent theoretical modeling has reported the piezoelectric behavior of the PTFE electret to be due to the presence of charges, the interaction of Maxwell stress, and deformation nonlinearity^{35,36}.

Comment 3.3: The results presented, for instance in Figure 4. includes a control where only ultrasound was used but no piezo materials. That is good, but it also requires controls WITH those materials present but without ultra sound.

Response: We thank the reviewer for the excellent suggestion. According to the reviewer's suggestion, control experiments with piezoelectric materials in the absence of ultrasound was investigated.

Fig. R8 MO removal by PTFE and PVDF, PE, TiO₂ without ultrasound irradiation. (This figure has been added into the revised Supplementary Information as new Supplementary Fig. 5)

Action: MO removal by PTFE and PVDF, PE, TiO₂ in the absence of ultrasound was investigated as shown in **Fig. R8**. We have provided corresponding discussion in the revised manuscript on Page 10:

In the absence of ultrasound, the above-mentioned catalysts displayed negligible MO removal, indicating the piezocatalytic reaction required ultrasonic stimulation (Supplementary Fig. 5).

Comment 3.4: The results shown in figure 4 for the bacteria are not clear - the photographs of the agar plates are not of high quality.

Response: Thanks for your suggestion.

Action: Higher quality photographs of the agar plates have been added to the revised manuscript in place of the previous low quality photographs.

Comment 3.5: Methods section: I found a lack of detail such as the geometry of the samples/containers. For instance in the bacteria experiments, the generation of ROS would have been at the membrane surface lining the beaker. How large was this beaker in diameter. We are told it was a 100 ml beaker but this provides little insight into the distance the ROS has to diffuse to react with the bacteria in suspension.

Response: We thank the reviewer for this insightful comment. Of note is that the ultrasonic wave is not uniformly distributed in the ultrasound bath likely due to use of discrete piezoelectric elements. We determined that the position of the beaker in the ultrasound cleaner slightly affected catalytic

performance. A photograph of the pollutant degradation experiments and bacterial disinfection experiments is displayed below (**Fig. R9**). The beaker is suspended within the ultrasonic bath such that the water level of the solution in the beaker is the same as the water level in the ultrasonic cleaner. The inner diameter of the beaker is about 4.8 cm and the height of water in the ultrasonic bath is about 11 cm.

Additionally, we agree with the reviewer that the ROS are generated at the membrane surface and since ROS typically have an ultrashort lifetime ($10^{-6}\sim 10^{-9}$ s) and diffusion distance (1~30 nm) in water (*Angew. Chem. Int. Ed.* 2019, 58(24), 8134-8138; *Trends Cell Biol.* 2008, 18(9), 443-450), they may only be present in the near PTFE surface environment. Therefore, ROS cannot effectively diffuse to the target bacteria or pollutant in solution. However, ultrasound drives rapid solution mixing, which can promote diffusion of bacterial and pollutant to the membrane surface to react with ROS or contact between homogenous ROS and the target species.

Fig. R9 Experimental setup photograph for pollutant degradation and disinfection of bacteria. (This figure has been added into the Supplementary Information as new Supplementary Fig. 10)

Action: To address this question from the reviewer, the following sentences were added into the revised manuscript on Page 16:

The inner diameter of the beaker is about 4.8 cm and the height of water in the ultrasonic bath is about 11 cm. The beaker is suspended within the ultrasonic bath such that the water level of the solution in the beaker is the same as the water level in the ultrasonic cleaner (Supplementary Fig. 10).

Comment 3.6: Overall, the results presented would be of high interest to many researchers interested in water remediation and biomedical treatment. There would appear to be many exciting applications, including pretreatment of feed water in membrane based water treatment plants.

Response: Many thanks for the reviewer's high affirmation of our research work.

REVIEWER COMMENTS

Reviewer #1 (Remarks to the Author):

Revised manuscript is improved significantly. Most of the comments are addressed scientifically and incorporated in the revised manuscript.

Reviewer #2 (Remarks to the Author):

Authors have almost replied to the questions. However, the following questions still not clear enough. Pls address it in detail.

1. Authors explain that “ ...the extreme ultrasonic cavitation pressure waves can cause massive PTFE deformation resulting in permanent structural defects. As a consequence, the concurrent electric fields can generate charges that can be trapped in the PTFE structural defects creating an electret state...

The authors explain the piezo effect of PTFE reported in Phys. Rev. E 90, 012603. If so, can we find the same effect in others polymer or inorganic materials? Can we observe the piezo-charges that were generated in nonpiezoelectric materials? Pls explain in detail.

2. Based on the explanation of the authors, the charges are not due to a non-centrosymmetric structure. The definition of the piezo-charges of PTFE should be clarified.

3. In the introduction section, the author should refer to non-polar polymeric electret more. (ex: How are they used in conventional electronic devices?) Some of the responses should be shown in the manuscript.

4. Is it possible to generate the same effect in other non-polar polymers? The non-polar polymers should be used as a control sample to show their degradation activities compared with PTFE.

5. A schematic diagram and semi-crystal PTFE structure should be expressed to show how structural defects are generated inside PTFE and how charges are trapped inside them during ultrasonic vibration.

Response to Reviewers' Comments

Reviewer #1 (Remarks to the Author)

Revised manuscript is improved significantly. Most of the comments are addressed scientifically and incorporated in the revised manuscript.

Response: We thank the reviewer again for their valuable suggestions and affirmation of our revised manuscript.

Reviewer #2 (Remarks to the Author)

Authors have almost replied to the questions. However, the following questions still not clear enough. Pls address it in detail.

Response:

We thank Reviewer 2 for the positive evaluation of our manuscript. We have revised our manuscript according to the new comments and the responses are detailed below.

General issues:

Comment 2.1: Authors explain that "...the extreme ultrasonic cavitation pressure waves can cause massive PTFE deformation resulting in permanent structural defects. As a consequence, the concurrent electric fields can generate charges that can be trapped in the PTFE structural defects creating an electret state..."

The authors explain the piezo effect of PTFE reported in Phys. Rev. E 90, 012603. If so, can we find the same effect in others polymer or inorganic materials? Can we observe the piezo-charges that were generated in nonpiezoelectric materials? Pls explain in detail.

Response 2.1: We appreciate the reviewer's thoughtful question. We investigated the sonocatalytic activities of other organic polymers (Polyethylene (PE), Polypropylene (PP)) and inorganic materials (TiO₂, Al₂O₃, MgO) under ultrasound irradiation. PE, TiO₂, Al₂O₃ and MgO have minimal catalytic activity for degradation of MO under ultrasound irradiation. The PP particles had measureable ultrasonic MO degradation (**Fig. R1**), but still significantly lesser than PTFE. Similar to PTFE, Polypropylene (PP) is also a non-polar polymer, which can be electrically polarized into electret. PP electrets are used in face masks for prevention of respiratory infections especially during the COVID-19 pandemic. The results in **Fig. R1** indicate that the degree of ultrasonic activation of various organic and inorganic polymers is highly dependent on their chemical composition and the underlying mechanism of their ultrasonic activation needs to be studied further to understand this phenomena.

In addition, the piezo-charges cannot be observed in nonpiezoelectric materials. The ultrasonic generation of ROS is due to the reaction of piezo-charges with water or oxygen. Since Al₂O₃ and MgO are nonpiezoelectric and thus have no piezo-charges, and have negligible sonocatalytic

activity for the degradation of MO.

Fig. R1 MO degradation as a function of time by piezocatalytic PTFE, PE, TiO_2 , and PVDF (a) and PP, Al_2O_3 and MgO under ultrasound irradiation for 1 h (b).

Comment 2.2: Based on the explanation of the authors, the charges are not due to a non-centrosymmetric structure. The definition of the piezo-charges of PTFE should be clarified.

Response 2.2: Many thanks for this comments. The charges in non-centrosymmetric materials are due to permanent electric dipoles. However, the charges in centrosymmetric electret, such as PTFE, are primarily a result of space-charges, and the charging of space-charge (or surface charge) electrets is usually achieved by injecting (or depositing) charge carriers via corona discharge, electrical breakdown radiation, light, pressure, or heat (Electrets (Topics in Applied Physics) (Springer-Verlag, Berlin, Heidelberg. New York, 1980)). In our study, ultrasonic irradiation of aqueous solutions generates at least three elements mentioned above, which individually or in combination may be the driving force behind the polarization of the non-polar PTFE electret. The ultrasonic PTFE structural deformation could create surface or bulk defects, which results in non-polar materials having permanent surface or space charges and apparent piezoelectricity.

Action: In the revised version of the manuscript, we added a description in the text of ultrasonic generation of PTFE piezo-charges on page 9 as displayed below. A schematic diagram of PTFE electret formation during ultrasound irradiation has also been added into Supplementary Information (Figure S5).

Meanwhile, the polarization charges in the piezoelectric PTFE electrets are primarily a result of

space or surface charges, and the creation of space-charge (or surface charge) electrets is achieved by injecting (or depositing) charge carriers via the high pressures, temperatures, and/or electric fields generated during ultrasound irradiation (**Supplementary Fig. 5**).

Fig. R2 Schematic diagram of PTFE electret formation during ultrasound irradiation (This figure has been added into the Supplementary Information as new Supplementary Fig. 5.)

Comment 2.3: In the introduction section, the author should refer to non-polar polymeric electret more. (ex: How are they used in conventional electronic devices?) Some of the responses should be shown in the manuscript.

Response 2.3: We thank the reviewer for this constructive comment.

Action: In the revised version of the manuscript, we added text on the applications of polymeric electrets in electronic devices on page 4 as shown in following paragraph, and cited the relevant references properly throughout the manuscript.

Non-polar polymer electret materials such as poly(tetrafluoroethylene) (PTFE; Teflon), polypropylene, and polystyrene are dielectrics that can quasi-permanently store charge or polarization. These organic electrets have been widely utilized in transducers (e.g. microphones and loudspeakers), electrophotography, electroactive air filters, and generators²⁴.

Comment 2.4: Is it possible to generate the same effect in other non-polar polymers? The non-polar polymers should be used as a control sample to show their degradation activities compared with PTFE.

Response 2.4: We thank the reviewer for raising this question, which we have partially answered in Response 2.1. Polyethylene (PE) is also non-polar polymer. In Fig. 4a, we investigated the sonocatalytic activity of PE (see below); however, we found that PE particles only yielded a sonocatalytic MO degradation rate constant of 0.057 h^{-1} that was similar to ultrasound alone (0.053 h^{-1}), and ~ 50 times lower than that of activated PTFE particles (2.81 h^{-1}).

Similar to PTFE, Polypropylene (PP) is another non-polar polymer that can be polarized into electret. PP electrets have been used in face masks for preventing respiratory infections, which have become popular during the COVID-19 pandemic (**Fig. R3a**). Here, we also investigated the catalytic degradation of MO by activated PP particles. The results indicated that the MO ultrasonic degradation with PP particles was greater as compared to ultrasound alone (**Fig. R3b**), but significantly lesser (6-fold) than with PTFE particles. Thus, a similar piezocatalytic effect is active in other non-polar polymers. However, as expected the degree of ultrasonic activation is polymer specific (PTFE \gg PP \gg PE), which is likely related to the polymer physical and chemistry properties e.g., the PTFE C-F bonds have a much greater polarity as compared to PP and PE C-H bonds. In summary, our discovery of PTFE piezocatalytic activity induced by ultrasonic irradiation here may only be the tip of the iceberg and the ultrasonic activation mechanism of other non-polar polymers should be further studied.

Fig. R3 Photography of face mask (a) and MO degradation by PTFE and PP under ultrasound irradiation for 1 h (b). $[\text{catalyst}]_0 = 0.25 \text{ g/L}$, $[\text{MO}]_0 = 5 \text{ mg/L}$.

Comment 2.5: A schematic diagram and semi-crystal PTFE structure should be expressed to show how structural defects are generated inside PTFE and how charges are trapped inside them during ultrasonic vibration.

Response 2.5: We thank the reviewer for this constructive suggestion. The non-activated PTFE as a whole is centrosymmetric, and the PTFE electret formation mechanism is primarily a result of the generation of space or surface charges during ultrasonic irradiation. Therefore, it is difficult to draw a specific polarized molecular structure change in PTFE (only C-F functionalization) similar to PVDF (half C-H and half C-F functionalization; **Fig. R4**).

The high pressures, temperatures, and electric fields generated by ultrasonic cavitation can cause large PTFE structural deformation to create permanent defects or voids (*Adv. Mater.* 2020, 32, 2000006). The air, oxygen, or water molecules in the voids can be ionized under the extreme ultrasonic irradiation conditions, and the positive and negative charges are deposited/injected at the ends of the voids or at the polymer surface (**Fig. R2**).

Fig. R4 Molecular dipole arrangement in PVDF. The α -phase has a *trans-gauche-trans-gauche* conformation (b), while the β -phase has a planar zigzag conformation (a). The α -phase PVDF, molecular dipoles are aligned in an opposite direction to each other, hence nonpolar, while in the β -phase the dipoles are aligned in such a fashion that self-cancellation does not occur; therefore the β -phase possesses a net dipole moment and hence is polar. (*J. Macromol. Sci. Part C: Pol. Rev.*, 31:4, 341-432)

Action: The schematic diagram of PTFE electret formation during ultrasound irradiation was added into the Supplementary Information.

Fig. R2 The schematic diagram of PTFE electret formation during ultrasound irradiation (This figure has been added into the Supplementary Information as new Supplementary Fig. 5.)

REVIEWERS' COMMENTS

Reviewer #2 (Remarks to the Author):

Authors have addressed all issues. The paper is highly recommended to publish in NC.

Response to Reviewers' Comments

Reviewer #2 (Remarks to the Author)

Authors have addressed all issues. The paper is highly recommended to publish in NC.

Response: We thank the reviewer again for their valuable suggestions and affirmation of our revised manuscript.